# Blood Microbiota Profile Is Associated with the Responsiveness of Postprandial Lipemia to *Platycodi radix* Beverage: A Randomized Controlled Trial in Healthy Subjects

**DOI:** 10.3390/nu15143267

**Published:** 2023-07-24

**Authors:** Seunghee Kang, Inhye Lee, Soo-yeon Park, Ji Yeon Kim, Youjin Kim, Jeong-Sook Choe, Oran Kwon

**Affiliations:** 1Logme Inc., Seoul 03182, Republic of Korea; 2Department of Nutritional Science and Food Management, Ewha Womans University, Seoul 03760, Republic of Korea; 3Department of Nutritional Science and Food Management, Graduate Program in System Health Science and Engineering, Ewha Womans University, Seoul 03760, Republic of Korea; 4Department of Food Science and Technology, Seoul National University of Science and Technology, 232 Gongneung-ro, Nowon-gu, Seoul 01811, Republic of Korea; 5Department of Agrofood Resources, National Institute of Agricultural Sciences, Rural Development Administration, Jeonbuk 55365, Republic of Korea

**Keywords:** association, blood microbiota profile, oral fat load, *Platycodi radix*, postprandial hyperlipemia, responsiveness

## Abstract

Prolonged postprandial hyperlipidemia may cause the development of cardiovascular diseases. This study explored whether postprandial triglyceride-rich lipoprotein (TRL) clearance responsiveness to *Platycodi radix* beverage (PR) is associated with changes in blood microbiota profiles. We conducted an 8-week randomized controlled clinical trial involving normolipidemic adults with low fruit and vegetable intakes. Participants underwent an oral fat tolerance test and 16S amplicon sequencing analysis of blood microbiota. Using the Qualitative Interaction Trees, we identified responders as those with higher baseline dietary fat intake (>38.5 g/day) and lipoprotein lipase levels (>150.6 ng/mL), who showed significant reductions in AUC for triglyceride (TG) and chylomicron-TG after the oral fat tolerance test. The LEfSe analysis showed differentially abundant blood microbiota between responders and non-responders. A penalized logistic regression algorithm was employed to predict the responsiveness to intervention on the TRL clearance based on the background characteristics, including the blood microbiome. Our findings suggest that PR intake can modulate postprandial TRL clearance in adults consuming higher fat intake over 38.5 g/day and low fruit and vegetable intake through shared links to systemic microbial signatures.

## 1. Introduction

Cardiovascular disease (CVD) is a leading cause of death globally, responsible for 32% of all-cause deaths, with 17.9 million people dying from CVD in 2019, according to the World Health Organization [1]. While reducing low-density lipoprotein (LDL) cholesterol remains the primary clinical target for lowering CVD risk [2], growing evidence supports the hypothesis that hypertriglyceridemia is also independently associated with an increased risk of cardiovascular events [3]. Additionally, postprandial abnormalities in triglyceride-rich lipoprotein (TRL) metabolism have emerged as clinically significant CVD risk factors [4]. Because most individuals consume meals throughout the day, strategies for modulating postprandial triglyceride (TG) are particularly relevant to current society. Postprandial TRLs consist of chylomicrons (CMs) of intestinal origin, very low-density lipoproteins (VLDLs) of hepatic origin, and their remnants [3,5]. They can enter the circulation through one of two metabolic pathways: an exogenous pathway for the transport of dietary lipids and an endogenous pathway for the metabolism of endogenously produced lipids [6]. If TRLs are not properly metabolized and accumulate in the circulation, they can penetrate the arterial wall and induce inflammatory/oxidative stress, thereby leading to CVD [3,7,8].

There is compelling evidence to suggest that specific foods or components can positively impact the magnitude and duration of circulating TRLs, ultimately reducing the risk of CVD [9]. *Platycodi radix*, the root of *Platycodon grandiflorum* A. DC., is a promising candidate for reducing circulating TRL levels [10]. This root contains various beneficial compounds, such as saponins, flavonoids, phenolic compounds, polyacetylenes, polysaccharides, and sterols [11]. Among these, platycosides, commonly known as saponins, stand out due to their bioactive properties that account for diverse biological effects [10]. Multiple types of platycosides have been isolated and studied, with platycoside E and platycodin D recognized as the primary triterpenoid saponins displaying pronounced biological activities [12]. Our previous study involving a single-dose study of *Platycodi radix* beverage (PR) conducted on healthy adult subjects demonstrated that PR effectively suppressed the postprandial lipemic response [13]. However, the precise mechanism by which PR impacts postprandial TRL metabolism in a repeated-dose clinical trial remains unclear.

The emergence of 16S rRNA sequencing and metagenomics techniques has provided novel opportunities to investigate the mutual interplay of gut microbiota, diet, and human disease [14], further supporting the idea that the gut microbiome plays a role in improving host metabolism [15]. Fecal sample collection, however, presents distinct challenges. The microbiome composition can be dynamic, influenced by factors such as diet and daily routines, which we could not standardize in our research. Consequently, pinpointing the ideal timing to obtain a representative glimpse of an individual’s microbiome was not straightforward. Additionally, gathering a sufficient volume for analysis posed significant difficulties when entire stool samples raised storage and transportation issues. Moreover, many people might find handling a fecal sample uncomfortable, distasteful, or embarrassing, leading to reluctance or improper collection techniques. Considering these factors, we opted against collecting fecal samples for our study. Concurrently, there is a growing interest in blood microbiota research concerning typical physiology, especially since Nikkari et al. [16] first discovered the presence of microbes in the blood of healthy individuals [17]. Blood microbiota analysis might seamlessly integrate with routine blood examinations, offering a deeper understanding of systemic health and disease.

Therefore, we hypothesized in this study that the effect of PR on postprandial TRL clearance might be related to changes in the blood microbiome. To test this hypothesis, we conducted an 8-week randomized clinical trial (RCT) comparing PR to a placebo and evaluated postprandial TRL clearance and blood microbiota profiles at baseline and endpoint. In addition, the Qualitative Interaction Trees (QUINT) algorithm was utilized to differentiate between responders and non-responders based on pre-treatment characteristics. At the same time, linear discriminant analysis (LDA) coupled with effect size (LEfSe) was used to identify differentially abundant blood microbiota in responders versus non-responders. Finally, a machine learning algorithm was employed to elucidate the contribution of background blood microbiota to TRL clearance responsiveness.

## 2. Materials and Methods

### 2.1. Test Material

ChunhoNcare Co., Ltd. (Busan, Republic of Korea) generously provided the placebo and PR. We have described the test material preparation procedure in our previous publication [13]. In brief, the process of extracting *Platycodi radix* involved pressurized hot water (in a 3:10 ratio) at 120 °C and 1.8–2 bar for 3 h. The extract was then concentrated to achieve a solid content of 6.5% by weight. After this, it was filtered and sterilized. The final beverage, packaged in 80 mL pouches, consisted entirely of this extract. The placebo beverage was designed to match the PR beverage in terms of taste, color, and flavor. It contained 2.3% flavoring agents and 97.75% purified water. The quality of PR was controlled to ensure it contained the optimum 8.4 ± 0.3° Brix and standardized to contain 35 mg/L of platycoside E, as determined by high-performance liquid chromatography (HPLC) equipped with an evaporative light scattering detector (Agilent Technologies, Palo Alto, CA, USA) and a Cadenza CD-C18 column (250 mm × 4.6 mm, 3 μm; Imtakt, Philadelphia, PA, USA). Figure 1 presents a representative chromatogram of PR. The placebo, packed in 80 mL pouches, contained 2.3% flavor components and 97.7% purified water, providing a similar taste, color, and flavor to PR.

### 2.2. Participants

We determined the required sample size based on our previous study, which found that a minimum of 96 participants (*n* = 48/group) was necessary to achieve a statistical power of 80% in postprandial TG responses, with a two-sided α-level of 0.05 [13]. To allow for a maximum dropout rate of 25%, we recruited 128 healthy adults over 20 years old, with fasting TG levels below 200 mg/dL, through online and poster advertisements.

Inclusion criteria required that participants had fasting TG levels below 200 mg/dL and were healthy adults over 20 years old. Exclusion criteria included the following: (1) habitually high fruit and vegetable intake, as determined by a recommended food score (RFS) > 36 (scale 0–47) [18]; (2) body mass index (BMI) ≥ 30 kg/m^2^ or < 20 kg/m^2^; (3) use of medication or dietary supplements in the past one month; (4) alcohol consumption ≥ 420 g/week; (5) smoking ≥ 20 cigarettes/day; (6) a history of body weight change ≥ 10% within the preceding six months; (7) exercising ≥ 10 h/week; (8) a history of hypersensitivity to test foods; (9) enrollment in any other clinical trials within the preceding one month; (10) pregnancy or breastfeeding; (11) difficulty in using a smartphone.

The protocol was approved by the Institutional Review Board of Ewha Womans University (IRB No. 148-19) and registered on the WHO’s International Clinical Trials Registry Platform (KCT0002861). In addition, written informed consent was obtained from each participant at screening.

### 2.3. Clinical Trial Design

Ninety-six eligible participants were randomly assigned to either the placebo or PR group using a computer-generated random block number table. The group allocation was blinded for both the investigators and participants. After a 2-week run-in period, participants were instructed to consume one pouch of PR or matched placebo beverage twice daily, providing a daily intake of 5.6 mg/day of platycoside E, for eight weeks. Compliance was measured by counting the number of unused pouches. Throughout the study, participants were advised to avoid high-fat diets and foods containing *P. radix*- or saponins. We assessed participants’ food intake using the RFS and meats, eggs, dairy, fried foods, fat in baked goods, convenience foods, fats added at the table, and snacks (MEDFICTS) questionnaires [19] at the baseline and end of the study.

Participants visited the research center in the morning at the beginning and end of the study. They consumed one pouch of PR or matched placebo along with a standardized high-fat shake consisting of 900 kcal, 58.9% from fat, 33.3% from carbohydrate, and 7.6% from protein [20] without water within 5 min. Blood samples were collected in K2 EDTA tubes (BD Vacutainer, Franklin Lakes, NJ, USA) at 0, 3, and 6 h after treatment, centrifuged at 1500× *g*, 4 °C for 15 min to isolate plasma, and stored at −80 °C until further analysis.

### 2.4. Biochemical Analyses

Plasma samples were collected at specific intervals after oral fat loading to measure biochemical markers. Total TG was assayed using a Hitachi 7600 automatic biochemical analyzer (Hitachi, Tokyo, Japan), and lipoprotein lipase (LPL) concentration was quantified using a commercially available ELISA kit (Cell Biolabs, San Diego, CA, USA). CM- and VLDL-TG were measured using an enzymatic colorimetric assay kit (TG-S, Asan Pharmacy, Hwasung, Republic of Korea) after isolation by salt density gradient ultracentrifugation (Beckman-Coulter, Fullerton, CA, USA) [21]. Fasting plasma glucose and insulin were measured by the Hitachi 7600 analyzer and Cobas 8000 e602 immunology analyzer (Roche, Basel, Switzerland), respectively. Homeostatic model assessment for insulin resistance (HOMA-IR) was calculated using the following formula: HOMA-IR = glucose (mg/dL) × insulin (uU/mL)/405.

### 2.5. DNA Extraction from Plasma Samples

DNA extraction and 16S rRNA sequencing were performed at MD Healthcare (Seoul, Republic of Korea). Briefly, fasting plasma samples were mixed with phosphate-buffered saline and centrifuged at 10,000× *g*, 4 °C for 1 min. The resulting supernatants were filtered through a 0.22 µm filter, boiled at 100 °C for 40 min, and then centrifuged at 20,000× *g*, 4 °C for 30 min. The microbiota DNA was extracted from the supernatants using a PowerSoil DNA Isolation Kit (MO Bio Lab., Carlsbad, CA, USA) and quantified with a QIAxpert system (QIAGEN, Hilden, Germany).

### 2.6. 16S Amplicon Sequencing and Taxonomic Assignments

PCR amplification was carried out using primers specific for the V3-V4 hypervariable regions of the 16S rDNA gene. The resulting amplicons were quantified, pooled at equimolar ratios, and sequenced using the MiSeq System Guide (Illumina, San Diego, CA, USA). The paired-end reads that overlapped were merged into longer reads covering the 16S rRNA V3-V4 region. Chimeric sequences were identified using VSEARCH against the SILVA gold database. The high-quality reads that were filtered and trimmed were clustered into operational taxonomic units (OTUs) using VSEARCH with a de novo clustering algorithm with a 97% sequence similarity threshold [22]. The representative sequences of each OTU were aligned to the SILVA 128 database using UCLUST under default parameters. The relative abundance of OTUs and microbial composition were calculated and analyzed for each group.

### 2.7. Statistical Analyses

An intention-to-treat analysis was conducted for clinical outcomes after verifying normal distribution using the Quantile to Quantile plot. Values that exceeded three times the interquartile range were identified as potential outliers and excluded from the study. The Student’s *t*-test was used for assessing continuous variables, and chi-squared tests were employed for categorical variables to determine baseline characteristics. Based on the trapezoidal rule, postprandial lipemic responses were calculated using the baseline-corrected area under the curve (AUC) [23]. A linear mixed-effect model (LMM) analysis with a random effect (participants), random error (within-participants), and fixed effects (treatment, period, and treatment × period interactions) was used to determine the effects of treatment. Qualitative treatment–subgroup interactions were analyzed using QUINT [24] in R with a QUINT package [25]. LMM analysis was performed to statistically compare the differences between responder and non-responder groups based on selected modifiers. A *p*-value < 0.05 was considered significant.

Relative abundance data determined alpha and beta diversities between the two treatment groups. LEfSe analysis was performed to identify differentially abundant blood microbiota using an alpha cut-off of 0.05 and an effect size cut-off of 3.0, followed by Spearman’s correlation between differential changes in clinical outcomes and featured genera in responders and non-responders. Furthermore, a logistic regression model with the least absolute shrinkage and selection operator (LASSO) penalty was constructed to determine the contribution of blood microbiome profiles at baseline to TRL clearance responsiveness. The model was validated using leave-one-out cross-validation (LOOCV) and its performance was evaluated by the receiver operating characteristic (ROC) analysis. Statistical analyses were performed using SAS version 9.4 and R version 3.6.1.

## 3. Results

### 3.1. Baseline Characteristics of the Participants

The participants’ flow at each trial stage is presented in the CONSORT diagram shown in Figure 2. Out of 96 eligible participants, 75 subjects (78%) completed the study, with 12 withdrawals from the placebo group and 9 from the PR group due to withdrawal of consent, loss of follow-up, medication, and syringe phobia. No significant adverse events were observed during the experiment, and most participants demonstrated good acceptance of the treatments, with compliance above 99%. The two groups had no significant differences in baseline characteristics, as shown in Table 1. The participants were identified as young (aged 30.0 ± 0.7 years) and overweight (BMI, 23.1 ± 0.2 kg/m^2^) adults with low antioxidant (RFS, 15.4 ± 0.6) and moderate fat (MEDFICTS score, 44.7 ± 1.5) intakes. However, the blood lipid parameters (fasting TG, total cholesterol, LDL cholesterol, and high-density lipoprotein cholesterol) were within the normal ranges.

### 3.2. Effects of PR on the Alterations in Postprandial TRL Clearance

We calculated the baseline-corrected AUCs at the baseline and end of the intervention to obtain single numbers expressing postprandial TRL clearances for three and six hours. Next, we used the LMM analysis to test the treatment–period interactions for postprandial lipemia parameters (Table 2, upper part). We found significant decreases in TG AUC_0→3h_ (β = −24.6, *p* = 0.025) and CM-TG AUC_0→3h_ (β = −68.1, *p* = 0.036) in the PR group compared with the placebo group. Although the trend in the results persisted when assessing the AUCs over 6 h, these parameters no longer held statistical significance. Specifically, the values for TG AUC_0→6h_ (β = −44.9, *p* = 0.132) and CM-TG AUC_0→6h_ (β = −99.7, *p* = 0.335) reflect this observation.

### 3.3. Responsiveness to Treatment on Postprandial TRL Clearance and Changes in Blood Microbiota Profiles

We performed a subgroup analysis to identify the modifier effect on postprandial TRL clearance using the baseline characteristics dataset, which includes anthropometric, biochemical, and lifestyle factors. The QUINT analysis for TG AUC_0→6h_ resulted in a pruned tree with three leaves (Figure 3a). Leaves 1 (*n* = 26) and 2 (*n* = 21), shown in red, represent the subgroup of non-responders for whom the treatment was ineffective. Leaf 3 (*n* = 49), shown in green, represents the subset of responders who benefit from treatment. We defined a responder as a subject with a greater LPL mass (150.6 ng/mL) and dietary fat intake (38.5 g/day) than the corresponding split levels at baseline. The subsequent LMM analysis in Figure 3b confirmed that responders showed significant improvements in AUC for TG (β = −98.9, *p* = 0.010) and CM-TG (β = −407.6, *p* = 0.005) from 0 to 6 h after the intervention with PR (blue dots) compared with the placebo (white dots). However, these parameters did not change or worsen due to the intervention in the non-responder group.

Next, we performed a taxonomic assignment of the 16S amplicon sequences in the blood of responders and non-responders. The bacterial DNA sequences found in blood mainly belonged to the Firmicutes (41.3%) and Proteobacteria (29.9%) phyla and, to a lesser extent, to the Bacteroidetes (13.3%) and Actinobacteria (12.7%) phyla (Figure 4a). The relative abundance of taxa at all levels is fully described in Appendix A. Alpha and beta analysis did not show significant differences due to the high variability of blood microbiome composition between individuals. However, the LEfSe algorithm analysis showed differentially abundant blood taxa between responders and non-responders (Figure 4b). The *Burkholderiaceae* (Family), *Ralstonia* (Genus), and *Prevotella* (Genus) were significantly enriched, while *Actinobacteria* (Phylum), *Comamonadaceae* (Family), *Tepidimonas* (Genus), *Pseudomonadaceae* (Family), *Pseudomonas* (Genus), and *Citrobacter* (Genus) were significantly diminished in responders compared to non-responders. Although Spearman’s correlation analysis revealed no significant association between differential taxa abundance and clinical outcomes, Figure 4c showed results in crossing regression lines of the plot between the responder and non-responder groups, except for *Comamonadaceae* (Family) and *Tepidimonas* (Genus), which were exceptional.

### 3.4. Predicting TRL Clearance Based on Background Characteristics

In order to investigate whether the background characteristics, including blood microbiome profile at the genus level, can predict the responsiveness to treatment on postprandial TRL clearance, we employed a machine learning algorithm. We used the logistic regression model with LASSO penalty and identified the best model which was a sum of 22 features that predicted the responsiveness of postprandial TRL clearance in response to PR intervention (Figure 5a). The model’s performance was excellent, with an ROC AUC of 0.989 (Figure 5b). Additionally, the box and scatter plot in Figure 5c demonstrated that the constructed model successfully predicted the responsiveness of TRL clearance to PR consumption (*p* < 0.0001).

## 4. Discussion

This study aimed to investigate the effects of PR consumption on postprandial TRL clearance and changes in blood microbiota profiles in an 8-week RCT of healthy adults over 20 years old. All group analyses showed limited improvements in postprandial TRL clearance in the PR group compared to the placebo group. However, subgroup analysis using the QUINT algorithm identified that individuals with higher LPL levels (150.6 ng/mL) and dietary fat intake (38.5 g/day) at baseline responded better to PR consumption, exhibiting significant improvements in postprandial TRL clearance compared to the placebo group. The LEfSe analysis identified 14 differentially abundant blood taxa between responders and non-responders. Furthermore, a machine learning algorithm trained on the background characteristics, including blood microbiome profile at the genus level, could predict individual responsiveness to intervention in postprandial TRL clearance, indicating a close association between blood microbiota profiles and intervention effectiveness.

Plasma TG concentration typically peaks at 3–5 h after a meal and returns to baseline within 6–8 h in normolipidemic subjects [26]. However, in our study, a standardized oral fat load providing 900 kcal (58.9% from fat) continuously evoked significant increases in plasma TRL concentration for up to 6 h, demonstrating an exaggerated and prolonged lipemic response in subjects with placebo treatment. This result parallels previous observations in normolipidemic adults who ingested a meal containing 30–80 g of fat [27,28]. Interestingly, the 8-week PR intervention had a limited impact on the lipemic responses and fasting clinical parameters compared to the placebo group. This limited finding prompted us to introduce a robust strategy for identifying the subgroups of subjects for whom the effectiveness of PR consumption differs from the others.

In the present study, we utilized an unsupervised machine learning algorithm, QUINT, to identify modifiers of postprandial TRL clearance response based on a range of variables measured at baseline [24]. RCTs often enroll narrowly defined subjects by applying strict inclusion/exclusion criteria. However, a high inter-individual variation in response to the interventions still exists, primarily due to the baseline characteristics of the study subjects [29]. Thus, identifying subgroups in clinical trials for whom the intervention may be more effective is interesting. Various methods have been developed to address this issue, including penalized regression models with interaction terms and tree-based partitioning. Among them, the QUINT method is designed to subdivide the RCT subjects by evaluating all possible binary splits for each independent variable using recursive partitioning [25]. The terminal nodes are then assigned to responders or non-responders for further analysis. In the current study, the QUINT method identified background dietary fat intake and LPL mass as effect modifiers for postprandial TRL clearance. Therefore, subjects with higher levels of these effect modifiers were classified as responders more likely to be influenced by PR intervention, showing complete suppression of TG and CM-TG AUCs. This approach might help improve the understanding of the health benefits of food sources, enabling personalized nutrition solutions [25].

Previous studies, including ours, have demonstrated the platycodins’ ability to inhibit pancreatic lipase, partially explaining its direct role in reducing dietary lipid digestion [13,30]. Pharmacokinetic studies using HPLC–MS/MS methods have shown that the platycodins have very poor oral bioavailability (<2%) in rats [31], suggesting thus they may impact lipid metabolism indirectly through alterations in gut microbiota composition. Building on this hypothesis, we employed a machine learning algorithm to investigate the contribution of gut-derived blood microbiota in predicting the responsiveness of the postprandial TRL clearance to PR consumption. In this study, we developed an interpretable and straightforward model using LASSO penalized logistic regression analysis [32]. Given the limited sources available for early-stage work, we utilized the LOOCV technique to select a model with improved classification performance since a separate dataset for model validation was not feasible in clinical investigation [33]. We selected the most parsimonious model by determining the regularization parameters among many possible models [34]. Our final model’s accuracy was 0.989, as determined by the area under the ROC, indicating excellent prediction performance [35].

The final model includes 22 features that represent dietary and clinical parameters, as well as a composite of blood microbiota at baseline. The dietary and clinical parameters encompass fat and energy intake, fasting LPL mass, VLDL, TG, and glucose levels. The blood microbiota belong to various phyla, such as Actinobacteria (*Corynebacterium* and *Rothia*), Firmicutes (*Clostridium sensu stricto 1*, *Finegoldia*, *[Eubacterium] coprostanoligenes group*, *Holdemanella*, *Intestinibacter*, *Lachnospiraceae NK4A136 group*, *Lachnospiraceae UCG-008*, *Lactobacillus*, *Ruminococcus 1*, *Ruminococcus 2*, and *[Ruminococcus] gnavus group*), and Proteobacteria (*Burkholderia-Paraburkholderia*, *Methylobacterium*, and *Pseudomonas*). Over the past few decades, studies have shown positive or negative associations between the abovementioned genera and lipid metabolism. Notably, both animal and human studies have reported on the role of Firmicutes and Actinobacteria in lipid metabolism and obesity [36]. However, previous studies have failed to provide consistent evidence supporting individual microbiota at the genus level as a reliable biomarker of lipid metabolism in the general human population under free-living conditions. This evidence supports the body of knowledge that discourages using individual bacterial taxa as biomarkers to predict the effects or targets of therapeutic agents.

In conclusion, our research suggests that the consumption of PR can be recommended as a beneficial dietary tool to enhance postprandial TRL clearance in normolipidemic adults with a fat intake of over 38.5 g/day and a low intake of fruit and vegetables. To support this suggestion, we utilized a QUINT algorithm to stratify the subjects into responders and non-responders for postprandial TRL clearance. Moreover, using the penalized logistic regression algorithm, we obtained further evidence to support the notion that changes in the blood microbiota profiles are associated with the responsiveness of postprandial TRL clearance to PR consumption. Our study is robust, as we evaluated a wide range of data using new technologies, such as omics and machine learning approaches. However, it is important to note the limitations of our study. Firstly, we obtained microbiota information based on blood, which is still in the early stages of analysis compared to fecal samples. Since different body regions may have distinct microbial communities [37], it was challenging to provide an in-depth discussion based on the most accumulated information on fecal microbiota. Secondly, while cross-validation is a reliable method, our prediction model needs to be externally validated. Lastly, although we identified platycoside E as a significant component in PR using the HPLC method, we did not perform additional experiments to determine the pharmacological activities of purified platycodins. The structure of platycoside E closely resembles that of platycodin D, with both being oleanane-type triterpenoid saponin. Platycoside E carries two additional glucose groups in comparison to platycodin D. Previous research suggests that platycoside E can be converted into platycodin D via the hydrolytic action of a de-glucosidase [38]. Furthermore, studies have demonstrated that purified platycodins possess pharmacological activities. This supports the notion that PR directly impacts the postprandial clearance of TRLs.

## Figures and Tables

**Figure 1 nutrients-15-03267-f001:**
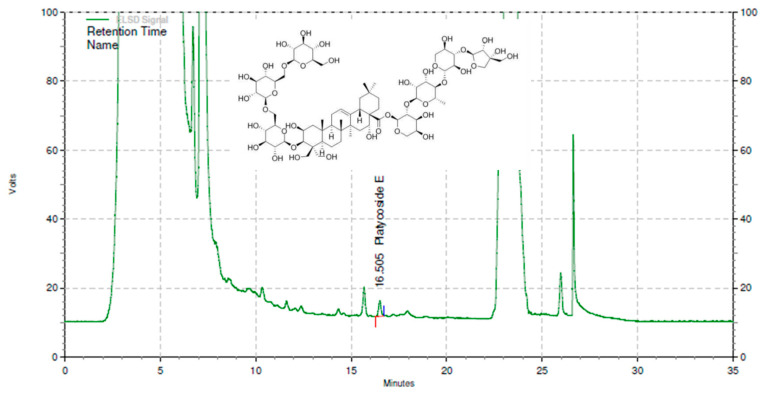
The HPLC chromatogram and chemical structure of platycoside E from *Platycodi radix* beverage. HPLC, high-performance liquid chromatography.

**Figure 2 nutrients-15-03267-f002:**
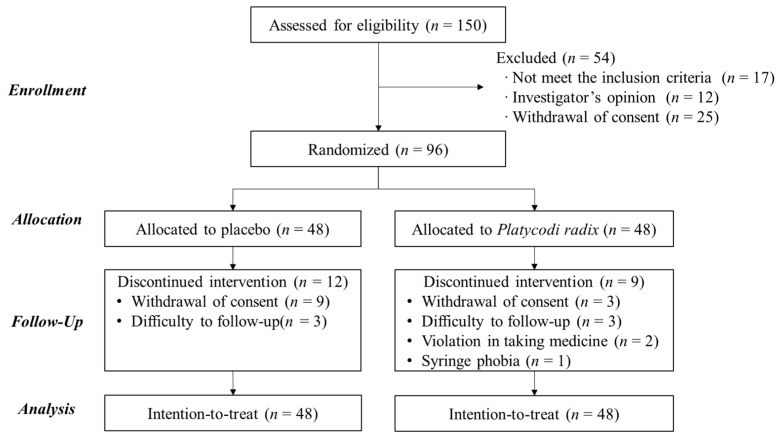
CONSORT diagram for the flow of all participants from recruitment to the end of the study.

**Figure 3 nutrients-15-03267-f003:**
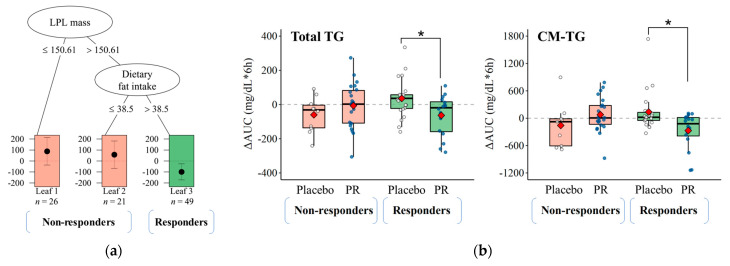
Partitioning tree and descriptive statistics of QUINT. (**a**) Partitioning tree with LPL mass (split point 150.61 ng/mL) and dietary fat intake (split point 38.5 g/day) as moderator variables. Leaf 1 and 2 (orange) represent non-responders, while leaf 3 (green) represents responders to the reduction of TG AUC after 8-week PR intake. (**b**) Descriptive statistics for TG and CM-TG AUCs, demonstrating that responders (green) benefit from PR treatment (blue dot) in contrast to placebo treatment (white dot). A linear mixed-effect model for treatment × period interaction terms was used to evaluate the *p*-values, and statistical significance is denoted by * (*p* < 0.05). The box and thick bar show the interquartile range and median, respectively, with the mean depicted by the red diamond. AUC, area under the curve; CM, chylomicron; LPL, lipoprotein lipase; PR, *Platycodi radix* beverage; QUINT, qualitative interaction trees; TG, triglyceride.

**Figure 4 nutrients-15-03267-f004:**
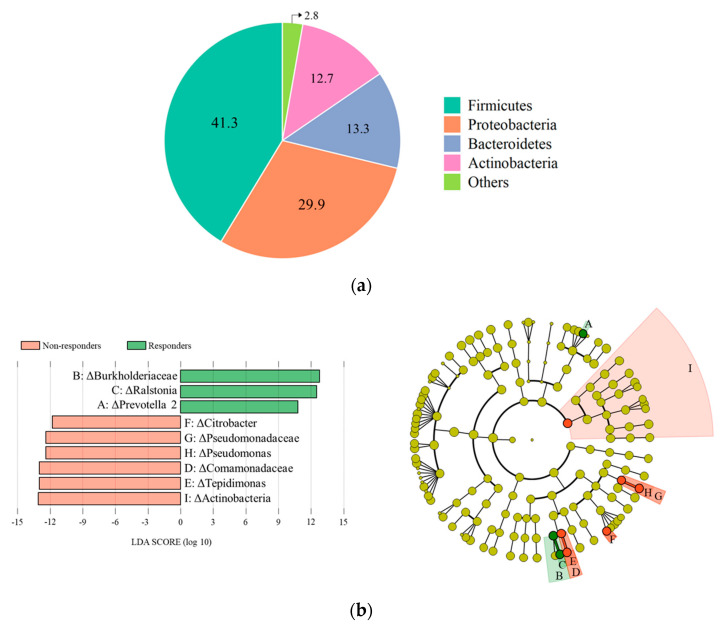
Differential abundance of blood microbiota between responder and non-responder groups after the 8-week intervention. (**a**) Pie chart representing dominant phyla composition (%) in plasma sample of all subjects. (**b**) LDA score histogram and LEfSe cladogram for individual taxa, highlighting the taxonomic differences between responders and non-responders. The orange and green highlighted taxa and nodes were significantly more abundant in the responder and non-responder groups, respectively. (**c**) Spearman’s correlation between the differential taxa abundance and clinical outcomes in each responder (green) and non-responder groups (orange). LDA, linear discriminant analysis; LEfSe, LDA effect size.

**Figure 5 nutrients-15-03267-f005:**
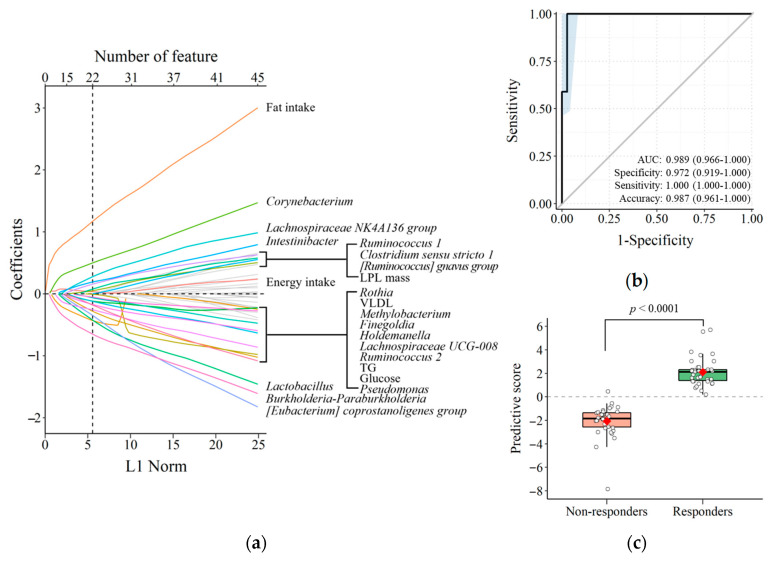
Background characteristics for predicting responsiveness of postprandial TRL clearance to PR intervention. (**a**) Visualization of the coefficient profiles for fitting the LASSO penalized logistic regression model. (**b**) ROC analysis of the prediction model. The diagonal line represents the reference line of 0.5. The light blue area represents the 95% confidence interval of the ROC curve estimate. (**c**) Calibration of the prediction model. A Student’s *t*-test determined the *p*-value. LASSO, least absolute shrinkage and selection operator; PR, *Platycodi radix* beverage; ROC, receiver operating characteristic; TRL, triglyceride-rich lipoprotein.

**Table 1 nutrients-15-03267-t001:** Baseline characteristics of participants in the intention-to-treat analysis.

Variables	Placebo (*n* = 48)	PR (*n* = 48)	*p*-Value ^1^
Age (years)	30.1 ± 1.6	29.8 ± 1.4	0.881
Sex (male/female, *n*)	18/30	18/30	1.000
Body weight (kg)	66.1 ± 1.5	63.0 ± 1.5	0.141
Height (cm)	167.4 ± 1.2	165.8 ± 1.4	0.381
Waist circumference (cm)	78.7 ± 1.1	76.2 ± 1.1	0.127
BMI (kg/m^2^)	23.5 ± 0.3	22.8 ± 0.3	0.159
SBP (mmHg)	117.3 ± 1.6	117.6 ± 1.6	0.867
DBP (mmHg)	72.1 ± 1.1	73.4 ± 1.2	0.432
Pulse rate (beats/min)	83.2 ± 1.5	84.7 ± 1.7	0.518
TG (mg/dL)	114.4 ± 11.0	102.8 ± 5.9	0.354
Total cholesterol (mg/dL)	187.4 ± 5.3	182.8 ± 4.5	0.507
LPL mass (ng/dL)	771.9 ± 257.5	635.5 ± 206.4	0.680
CM (mg/dL)	155.0 ± 31.5	125.5 ± 21.2	0.439
VLDL (mg/dL)	36.2 ± 4.3	43.0 ± 4.2	0.258
LDL (mg/dL)	116.1 ± 4.4	113.6 ± 4.3	0.686
HDL (mg/dL)	66.1 ± 2.4	65.9 ± 2.4	0.951
Smoker (*n*, %)	7 (7.29)	10 (10.42)	0.423
Alcohol drinker (*n*, %)	29 (30.21)	26 (27.08)	0.536
Physical activity (MET min/week)	1703.2 ± 228.0	1751.0 ± 298.5	0.899
Dietary fat intake (g/day)	46.0 ± 2.4	46.8 ± 2.0	0.809
RFS	16.0 ± 1.1	15.3 ± 1.0	0.610
MEDFICTS	50.0 ± 3.1	45.8 ± 2.5	0.291

Values are presented as mean ± SE or *n*. BMI, body mass index; CM, chylomicron; DBP, diastolic blood pressure; HDL, high-density lipoprotein; LDL, low-density lipoprotein; LPL, lipoprotein lipase; MEDFICTS, meats, eggs, dairy, fried foods, fat in baked goods, convenience foods, fats added at the table, and snacks; PR, *Platycodi radix* beverage; RFS, recommended food score; SBP, systolic blood pressure; TG, triglyceride; VLDL, very low-density lipoprotein. ^1^ The Student’s *t*-test for continuous variables and the chi-square test for categorical variables were used to compare the treatments.

**Table 2 nutrients-15-03267-t002:** Comparison of postprandial lipemic responses to an oral fat load at baseline and endpoint in the placebo and PR groups.

Variables	Placebo				PR				β ^2^	*p*-Value
	Baseline (*n* = 48)	Endpoint (*n* = 36)	β ^1^	*p*-Value	Baseline (*n* = 48)	Endpoint (*n* = 39)	β	*p*-Value		
Total TG (mg/dL × 3 h)	28.0 ± 6.6	40.1 ± 8.7	12.1	0.126	43.6 ± 6.4	31.1 ± 8.4	−12.4	0.098	−24.6	0.025
CM-TG (mg/dL × 3 h)	44.8 ± 16.0	63.4 ± 19.8	18.6	0.431	91.4 ± 15.3	41.9 ± 18.2	−49.5	0.025	−68.1	0.036
VLDL-TG (mg/dL × 3 h)	16.2 ± 3.3	12.5 ± 3.4	−3.7	0.415	9.5 ± 3.2	8.1 ± 3.2	−1.4	0.747	2.3	0.716
LPL mass (ng/mL × 3 h)	39.5 ± 15.4	28.3 ± 15.7	−11.1	0.451	32.9 ± 15.0	44.4 ± 15.1	11.5	0.418	22.6	0.270
Total TG (mg/dL × 6 h)	171.1 ± 20.0	185.0 ± 25.5	13.9	0.520	208.0 ± 19.6	177.0 ± 24.7	−31.1	0.130	−44.9	0.132
CM-TG (mg/dL × 6 h)	334.6 ± 60.8	357.4 ± 66.2	22.8	0.763	387.7 ± 58.3	310.9 ± 61.0	−76.8	0.274	−99.7	0.335
VLDL-TG (mg/dL × 6 h)	51.0 ± 8.8	51.6 ± 11.1	0.6	0.965	33.6 ± 8.7	32.4 ± 10.5	−1.2	0.928	−1.8	0.925
LPL mass (ng/mL × 6 h)	98.6 ± 43.0	105.1 ± 42.7	6.6	0.877	105.5 ± 42.0	128.5 ± 41.2	23.0	0.573	16.5	0.779
BMI (kg/m^2^)	23.5 ± 0.3	23.5 ± 0.3	−0.03	0.758	22.8 ± 0.3	22.8 ± 0.3	−0.01	0.906	0.02	0.888
Fasting glucose (mg/dL)	86.2 ± 1.2	87.1 ± 1.1	0.86	0.367	87.2 ± 1.2	85.7 ± 1.0	−1.46	0.111	−2.32	0.081
Fasting insulin (uU/mL)	7.44 ± 0.53	7.37 ± 0.60	−0.08	0.877	7.18 ± 0.53	7.04 ± 0.59	−0.14	0.775	−0.06	0.931
HOMA-IR	1.64 ± 0.12	1.62 ± 0.14	−0.02	0.878	1.56 ± 0.12	1.50 ± 0.13	−0.06	0.585	−0.04	0.787

Values are presented as adjusted least square mean ± SE. BMI, body mass index; CM, chylomicron; HOMA-IR, homeostatic model assessment for insulin resistance; h, hour; LPL, lipoprotein lipase; PR, *Platycodi radix* beverage; TG, triglyceride; VLDL, very low-density lipoprotein. ^1^ The β estimates and *p*-values were obtained using a linear mixed-effect model within the group between baseline and endpoint. ^2^ The β estimates and *p*-values were obtained using a linear mixed-effect model for treatment × period interaction term.

## Data Availability

Data sharing is not applicable.

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
