# Peer review of "Blood Microbiota Profile Is Associated with the Responsiveness of Postprandial Lipemia to Platycodi radix Beverage: A Randomized Controlled Trial in Healthy Subjects"

_nutrients, 2023, doi:10.3390/nu15143267_

Round 1

Reviewer 1 Report

Dear Authors,

In the one of the last supplements to the European Pharmacopoeia (Supl. 9.4) figures several herbal medicinal raw materials, so far unknown and not used in official Central European phytotherapy. Platycodon grandiflorus (balloon flower) provides the new saponin raw material – Platycodi radix. For about 20 years, this plant has had the World Health Organization (WHO) monograph. So far, about 70 triterpene saponins have been isolated from P. grandiflorus.  

The article presents the results of a study of the PR effects on postprandial TRL clearance. The research methods are described in detail with reference to previous studies. The discussion of Author’s own results with the involvement of data from other studies was carried out in a very interesting and qualified manner. A possible explanation of the pathophysiological mechanisms of the changes is given.

However, there are some comments and questions to be cleared for an improvement of the article. I have added minor comments in the text.

It was only in the last paragraph of the text that I got an answer to the question why the authors highlight platicoside E as potentially beneficial. It's known that the most important, biologically active ingredients are triterpene saponins contained in the roots of P. grandiflorus - mainly pentacyclic compounds of the oleanane type, including platicodin D and polygalacin D. Other compounds include: flavonoids (platiconin, taxifolin, platicoside, apigenin, luteolin), phenolic compounds, polyacetylenes, polysaccharides, sterols. This should have been mentioned in the "Introduction". Saponins inhibit the absorption of fats from food. They also counteract free radicals - there are studies confirming such an effect of platicoside E.

Author Response

Thank you for valuable comments. Please see the attachment.

Dear Reviewer,

We have revised the manuscript (nutrients-2512253) entitled "Blood Microbiota Profile is Associated with the Responsiveness of Postprandial Lipemia to Platycodi Radix Beverage: A Randomized Controlled Trial in Healthy Subjects." We have endeavored to address each concern raised by the reviewers and explained all subsequent changes to the manuscript in this 'response to reviewers' file. And the corrections have been highlighted in red in the clean version of the manuscript. We think the changes motivated by your comments have vastly improved the paper.

  1. It was only in the last paragraph of the text that I got an answer to the question why the authors highlight platicoside E as potentially beneficial. It's known that the most important, biologically active ingredients are triterpene saponins contained in the roots of P. grandiflorus - mainly pentacyclic compounds of the oleanane type, including platicodin D and polygalacin D. Other compounds include: flavonoids (platiconin, taxifolin, platicoside, apigenin, luteolin), phenolic compounds, polyacetylenes, polysaccharides, sterols. This should have been mentioned in the "Introduction". Saponins inhibit the absorption of fats from food. They also counteract free radicals - there are studies confirming such an effect of platicoside E.:

Line 81: Why platycoside E? So far, about 70 triterpene saponins have been isolated from P. grandiflorus.

Line376-380: It was only at this point that I got an answer to the question why the authors emphasize platicoside E as potentially beneficial?

  • Thank you for pointing out this issue. Following the reviewer's comment, we have added more explanation why we highlight platycoside E as potentially beneficial in the Introduction (Lines 55-61) and Discussion (Lines 398-404) sections.

  1. Line125: the space is not needed.
  • Corrected (Line 147). Thank you.

  1. Line206: In my opinion, it cannot be said that "statistical significance was diminished". It is clearly visible that after 6 hours the reduction of the mentioned parameters is no longer statistically significant.
  • Thank you for the insightful comment. In response to your comment, we have rephrased the relevant sentences, which read as follows (Line 226-229): “Although the trend in the results persisted when assessing the AUCs over 6 hours, these parameters no longer held statistical significance. Specifically, the values for TG AUC06h (b = -44.9, p = 0.132) and CM-TG AUC06h (b = -99.7, p = 0.335) reflect this observation.”

  1. Table 1. p-value for sex: Really? I would write ">0.999"
  • Because the data are discrete and the mean difference was exactly 0, we could get an exact p-value of 1 on a Chi-square test.

  1. Line 238-240: According to the International Code of Nomenclature of Prokaryotes, the scientific names of taxa of all ranks should be in italics (correctly recorded on page 11).
  • Corrected as commented (Lines 260-267). Thank you.

Reviewer 2 Report

This work investigated the relationship between Platycodi Radix Beverage (PR) and postprandial triglyceride-rich lipoprotein (TRL) clearance responses at the blood microbiota level. The findings suggest that PR intake can modulate postprandial TRL clearance in adults with a daily intake of more than 38.5 grams of fat and low fruits and vegetables through co-association with systemic microbial profile, which is certainly of concern. However, I request the following clarifications or modification.

1. Format issues.

(1) Writing errors. Line 78, the company's expression format is incorrect. Line 124, the expression of Unit of time is incorrect and should be singular.

(2) The format of Unit of time is not uniform. The abbreviation is used in Section 2.3, and the full spelling is used in Section 2.5.

(3) The format of Unit of temperature is not uniform. For example, Line 143, whether to keep space between numeral and symbol should be consistent.

(4) The use of symbols is inconsistent in full text. For example, in section 2.6 and the Discussion section, there are three different forms of mixed use: “-” “~” “−”.

(5) Inconsistent capitalization of text, such as writing "Figure 3A" on Line 221, when the actual text in the image is “(a)”.

2. As the study mentioned, a taxonomic assignment of the 16S amplicon sequences in the blood of responders and non-responders was performed. Why is there no complete display of the blood microbiology level ratio of responders and non-responders in Phylum, Family and Genus levels? As shown in Figure 4, nine indicators are discussed, why is Actinobacteria the only one with a specific numeral (12.7%), and the remaining eight are not?

3. Why is there no ingredient list and percentage content of each ingredient in the drink, and if there are other ingredients, why only platycoside E was selected for the study? How can the correlation between content and biological activity be illustrated? Could other low-content ingredients play the same role in this study?

4. Will platycoside E be affected by cytochrome P450 enzymes? Will this cover up or exaggerate its biological activity?

5. As mentioned in the article, the study of blood microbiota is still in its early stages, and the gut is the major system for absorption, why not include intestinal microbiota research in the study?

Author Response

Dear Reviewer,

We have revised the manuscript (nutrients-2512253) entitled "Blood Microbiota Profile is Associated with the Responsiveness of Postprandial Lipemia to Platycodi Radix Beverage: A Randomized Controlled Trial in Healthy Subjects." We have endeavored to address each concern raised by the reviewers and explained all subsequent changes to the manuscript in this 'response to reviewers' file. And the corrections have been highlighted in red in the clean version of the manuscript. We think the changes motivated by your comments have vastly improved the paper.

1-1. Format issues

Writing errors. Line 78, the company's expression format is incorrect. Line 124, the expression of Unit of time is incorrect and should be singular.

  • Corrected as commented (Lines 94 and 146). Thank you.

1-2. Format issues

The format of Unit of time is not uniform. The abbreviation is used in Section 2.3, and the full spelling is used in Section 2.5.

  • Corrected as commented (Lines 145 and 147). Thank you.

1-3. Format issues

The format of Unit of temperature is not uniform. For example, Line 143, whether to keep space between numeral and symbol should be consistent.

  • Corrected as commented (Line 165). Thank you.

1-4. Format issues

The use of symbols is inconsistent in full text. For example, in section 2.6 and the Discussion section, there are three different forms of mixed use: "-" "~" "−".

  • As per your suggestions, we have made corrections in 2.6 and Discussion sections. When denoting a range of numbers, we have employed the en dash. In other contexts, we have standardized to use of a regular dash. We appreciate your input.

1-5. Format issues

Inconsistent capitalization of text, such as writing "Figure 3A" on Line 221, when the actual text in the image is "(a)".

  • Corrected as commented (Lines 243, 248, 255, 260, 265, 299 and 300). Thank you.

  1. As the study mentioned, a taxonomic assignment of the 16S amplicon sequences in the blood of responders and non-responders was performed. Why is there no complete display of the blood microbiology level ratio of responders and non-responders in Phylum, Family and Genus levels? As shown in Figure 4, nine indicators are discussed, why is Actinobacteria the only one with a specific numeral (12.7%), and the remaining eight are not?
  • We have provided comprehensive details regarding the blood microbiology level ratios for both responders and non-responders at the Phylum, Family, and Genus levels in Supplementary Table S1. This table also encompasses the data for the remaining eight taxa (Lines 256 and 405-407).

  1. Why is there no ingredient list and percentage content of each ingredient in the drink, and if there are other ingredients, why only platycoside E was selected for the study? How can the correlation between content and biological activity be illustrated? Could other low-content ingredients play the same role in this study?
  • In 2.1. Test Material section (Lines 95-101), we have provided a succinct overview of the production processes for both the test and placebo beverages. The test beverage is solely derived from the Platycodi radix extract, leading us to choose platycoside E as a marker compound for standardization.
  • Our test material, PR, is not a singular purified compound but an extract encompassing various chemical components. As underscored in the discussion section (Lines 395-398), one of the limitations of this study is the absence of further experiments to determine the pharmacological actions of isolated platycosides.

  1. Will platycoside E be affected by cytochrome P450 enzymes? Will this cover up or exaggerate its biological activity?
  • The cytochrome P450 enzymes are pivotal in the body's metabolism of xenobiotics, and their activity can be modulated by various compounds, acting either as inhibitors or inducers.
  • While one might hypothesize that platycosides could impact the cytochrome P450 enzymes, our study did not delve into this dimension. Moreover, to the best of our knowledge, there is a dearth of detailed insights in the current literature concerning the effects of platycosides on these enzymes.
  • Nonetheless, it is worth highlighting that existing research underscores the central role of cytochrome P450 in the biosynthetic pathways of saponins, especially in the hydroxylation or oxidation processes of triterpene structures.

  1. As mentioned in the article, the study of blood microbiota is still in its early stages, and the gut is the major system for absorption, why not include intestinal microbiota research in the study?
  • We have provided the reason why we did not include the analysis of fecal microbiome analysis but used blood microbiota research in the Introduction section (Lines 69-81).
  • Collecting fecal samples for microbiome analysis was not straightforward due to the fluctuation of microbiome composition by diet and daily routines. Therefore, timing the collection to capture a "typical" snapshot of an individual's microbiome can be challenging. Collecting the entire stool sample is difficult due to storage and transportation issues. In addition, collecting a fecal sample is uncomfortable, distasteful, or embarrassing to many people, leading to hesitancy or improper collection techniques. Given these challenges, we did not collect fecal samples for this study.   

Round 2

Reviewer 2 Report

All comments were addressed in the revised manuscript.